# Unexpected Slow Kinetics of Poly(Methacrylic Acid) Phase Separation in the Semi-Dilute Regime

**DOI:** 10.3390/polym14214708

**Published:** 2022-11-03

**Authors:** Clément Robin, Cédric Lorthioir, Abdoulaye Fall, Guillaume Ovarlez, Catherine Amiel, Clémence Le Coeur

**Affiliations:** 1Institut Chimie et Materiaux Paris Est, Université Paris Est Créteil, CNRS, 2 Rue Henri Dunant, 94320 Thiais, France; 2Laboratoire de Chimie de la Matiere Condensee de Paris, Sorbonne Universite, Cnrs, College de France, 4 Place Jussieu, 75005 Paris, France; 3Laboratoire Navier, UMR 8205-Université Gustave Eiffel, Ecole des Ponts, CNRS, 77420 Champs sur Marne, France; 4LOF UMR 5258 (CNRS-Solvay-Université de Bordeaux) 178, Avenue du Dr Schweitzer, 33608 Pessac, France; 5Laboratoire Leon Brillouin, Cea-Cnrs (Umr-12), Cea Saclay, Universite Paris-Saclay, 91191 Gif-Sur-Yvette, France

**Keywords:** poly(methacrylic-acid), LCST polymer, SANS

## Abstract

Poly (methacrylic acid) (PMAA) solutions are known to exhibit a lower critical solution temperature (LCST). A temperature-composition phase diagram of PMAA has been constructed by standard cloud point determination through transmittance measurements, and also by studying the steady states reached under phase separation. This allows us to reconstruct the binodal curve describing the phase behavior of PMAA for both low and high concentration regimes, and to determine accurately the LCST temperature. In a second step, the structures formed following a temperature jump above the cloud point and their evolution in time have been investigated at the nanoscale using small angle neutron scattering (SANS). This approach shows that the formation of phase-separated nanostructures is a slow process, requiring more than 12 h. The formed structures are then shown to depend on the amplitude of the temperature jump above the cloud point. An original mechanism of phase separation is identified in the semi-dilute regime. The growth of micrometric-size droplets with an inner structure displaying the rheological properties of a gel leads to the formation of a percolating network which hinders the influence of gravity. Such a result can explain the slow kinetics of the PMAA LCST transition.

## 1. Introduction

In the past few years, stimuli-responsive (temperature, pH, ionic strength, pressure) polymers have drawn considerable interest due to their potential for a wide range of applications. Such polymers can be used in biomedical applications such as on-demand drug delivery [1,2,3,4,5,6,7], diagnostics [4], bioadhesion [8,9,10,11,12] or tissue engineering [3,13,14] due to their thermally reversible coil-to-globule transition. Polymers with a thermoresponsive behavior may display either an upper (UCST) or a lower (LCST) critical solution temperature. Among the LCST polymers, the structure as well as the thermal properties of poly(*N*-isopropylacrylamide) (PNIPAm) have been extensively investigated [15,16,17,18,19,20,21] and this latter is now used in many materials mainly related to biological applications [22,23,24,25,26].

Poly (methacrylic acid) (PMAA) is a thermo-responsive polyelectrolyte which exhibits a LCST behavior as reported for the first time by Eliassaf and Siberberg [27]. This polymer has been far less investigated than PNIPAm even though the chemical structure of the PMAA repeat unit results in specific and interesting properties in water. In particular, the mechanism underlying the LCST phase separation for PMAA significantly differs from that involved in PNIPAm. From another point of view, although the chemical structure of the PMAA repeat units is rather close to that of poly(acrylic acid) (PAA), their behavior in water is drastically different, since PAA is a UCST polymer [28,29,30,31]. This specificity of PMAA is generally assigned to the methyl group located in the alpha position of the carboxylic acid function. As for any polyacid, the PMAA conformation depends on pH and ionic strength. At a low degree of ionization in water, synergetic effects resulting from both hydrophobic interactions, related to the presence of methyl groups, and intramolecular hydrogen bonds between acidic groups are responsible for the hypercoiled conformation of the PMAA chains. Nevertheless, above a critical ionization degree, PMAA exhibits an extended chain conformation is a way similar to a conventional highly charged polyelectrolyte. The conformation of PMAA has been studied by various experimental approaches such as viscosimetry [32], dynamic light scattering (DLS) [33], potentiometric titration [34,35], calorimetry [34], Small-Angle Neutron Scattering (SANS) [36,37] and Small-Angle X-ray Scattering (SAXS) [37,38], especially at low pH, for which PMAA displays a LCST behavior. Poly(methacrylic acid) exhibits a conformational transition with pH, from compact coil at low pH to extended coil at pH larger than pKa°(pKa° = 5 [38]) as expected for polyelectrolytes [33,39,40,41,42]. Small-scale rearrangements occur between pH = 2 and pKa°. Specifically, for PMAA, the nature of the compact conformation at low pH was not completely elucidated. Some studies suggest that the polymer coil has overall dimensions intermediate between those of a compact sphere without water molecules inside it and a Gaussian coil with a statistical segment close to that of other vinyl polymers. It has been described as an hypercoiled structure [23,37,38,43].

Although PMAA is often cited among the LCST polymers, phase diagrams or cloud point determination are scarcely found in the literature. Its thermosensitive behavior has recently been studied in an organic solvent (1,2-dimethoxyethane) [44] which evidenced that a temperature increase induces the formation of PMAA aggregates which are favored by cyclic H-bonded dicarboxylic dimers. These results are consistent with some results found in aqueous solutions and in particular with the fact that PMAA exhibits a thermosensitive behavior only at low pH [39]. The balance between the moderate hydrophobicity of the methyl groups and the hydrophilic nature of the carboxylic acid functions is believed to be responsible for this behavior. A weak change in the degree of ionization of the PMAA chain enhances the hydrophilicity and hinders its thermosensitive behavior. As mentioned by Sedlàk [39], the addition of HCl decreases the solvent quality of water by suppressing self-dissociation of the PMAA chains. Such a reduction is significant since the LCST falls from 73.8 °C in H_2_O with 20 mM of HCl to 38.8 °C with 1 M of HCl. In this contribution, the shrinking of the PMAA chains upon increasing the temperature has been observed at low concentration (c = 0.5 g/L) by light scattering. More generally and to the best of our knowledge, the LCST behavior of PMAA has been mostly studied in the dilute regime. The nature of the aggregates formed in both semi-dilute and concentrated regimes has not yet been elucidated, despite the fundamental and application relevance of this topic. 

The structures formed in the semi-dilute and concentrated regimes above the cloud point depend on both polymer concentration and temperature at which the LCST phase separation is carried out [45,46]. In the present work, the phase diagram of a PMAA/water system is determined using cloud point measurements and also by studying the steady states achieved after phase separation. Based on these results, Small-Angle Neutron Scattering (SANS) was used to investigate the structures formed at the nanoscale above the cloud point and their evolution in time.

## 2. Experimental Section

### 2.1. Materials

Poly(methacrylic acid) (PMAA) was purchased from Polymer Science Inc. The molar mass and molar mass distribution were obtained using size-exclusion chromatography (SEC). Prior to analysis, the polymer was purposely modified by methylation of the PMAA carboxylic acid groups using trimethylsilyldiazomethane [46]. The samples were analyzed in THF at a concentration of 10 g/L at 25 °C. The column used was a polystyrene mixed C13 column for organic solvents. The setup was equipped with a refractive index detector (RI) operating at 658 nm and a light scattering detector. The average molar masses (number-average molar mass M_n_ and mass-average molar mass *M_w_*) and the polydispersity (PDI) value were derived from the RI signal using a calibration curve obtained with poly(methyl methacrylate) standards purchased from Polymer Laboratories. The radius of gyration of the polymer was determined using a small-angle X-ray scattering (SAXS) experiment on a salt-free PMAA aqueous solution at 20 g/L. The analysis of the SAXS data was carried out using the standard Zimm formula assuming a Debye form factor including a Schulz–Zimm distribution [38]. Considering the average molar mass (*M_w_*) determined from SEC measurements and the radius of gyration obtained from SAXS measurements, one can deduce an estimate of *c**, the critical overlap concentration using the following formula:(1)c∗=Mw43πNARg3
where *N_A_* is the Avogadro number. The result obtained for *c**, 23 g/L (which corresponds to a volume fraction Φ_PMAA_ * = 0.0178), is consistent with that already obtained by viscosimetry on the same homopolymer [47]. All the PMMA characteristics that we have obtained are summarized in Table 1. 

Different samples with concentrations from 5 to 140 g/L were prepared. The solutions of neutral PMAA were prepared by dissolving the dry polymer powder in D_2_O (99 atom % D) purchased from Eurisotop (France). Solutions were stirred using a rotational shaker at a very low speed during 1 week prior to any further measurements. 

### 2.2. Methods

#### 2.2.1. Cloud Point Measurements

Samples were filled into 1 cm path length quartz cells. Light transmission at a wavelength of λ = 600 nm was measured during a 1 °C/min heating ramp starting from room temperature using a HP Agilent 8453 UV–visible spectrophotometer equipped with a Peltier temperature controller. Samples were homogenized within the cell using a magnetic stirrer. The cloud point temperature of each sample was determined as the temperature at which 50% transmittance is observed. In this study, all the samples have been prepared in D_2_O in order to correlate directly the cloud point measurements with the SANS experiments.

#### 2.2.2. Determination of the Phase Diagrams from Phase Separation

PMAA solutions in D_2_O have been prepared at different volume fractions, (Φ_PMAA_)_0,_ and kept at a fixed temperature above the cloud point (1 °C above or 10 °C above depending on the samples) for 3 weeks. After such a time interval, a stationary state is reached, and a phase separation has occurred with a polymer-rich phase at the bottom of the vial and a polymer-poor phase in the supernatant. Differences between the densities of the two phases enable the supernatant to be separated from the rest of the solution. The volume of each phase (V_dil,_ for the dilute–polymer-poor–phase, and V_conc,_ for the concentrated–polymer-rich–phase) was determined by weighing and by using the densities derived for each phase (see below). The volume fractions of both polymer-rich phase (*x_conc_* = V_conc_/V_tot_) and polymer-poor phase (x_dil_ = V_dil_/V_tot_) in the sample were then calculated. Each phase was successively weighed before and after oven-drying so that the corresponding weight fraction of PMAA could be determined. Then, these weight fractions were converted into volume fractions (*Φ_PMAA,conc_* for the polymer-rich phase and *Φ_PMAA,dil_* for the polymer-poor phase) using the density values of bulk PMAA (ρ_PMAA_ = 1.29) and D_2_O (ρ_D2O_ = 1.104). This allows us to provide a phase diagram of PMAA in D_2_O as schematized on Figure 1. 

#### 2.2.3. Small-Angle Neutron Scattering (SANS)

SANS experiments were performed on the beamline PACE at Laboratoire Léon Brillouin, CEA Saclay using 1 mm Helma cells. For measurements below the cloud points, in solutions at 25 °C, 3 configurations were used: the SANS incident wavelength and Detector-Sample-Distance (DSD) were λ_0_ = 5 Å and DSD = 3 m or 1 m on the one hand or λ_0_ = 12 Å and DSD = 4.6 m on the other hand. These configurations enable coverage of a scattering wavevector q range extending from 0.003 Å^−1^ to 0.3 Å^−1^.

Above the cloud point, time-resolved SANS measurements were performed. In order to monitor the kinetics of the phase separation, only two configurations were considered: λ_0_ = 5 Å or 12 Å and DSD = 3 m. During the phase separation, a sequence of measurements on a given sample consists of 4 measurements during 10 min at λ_0_ = 12 Å followed by a single measurement during 10 min at λ_0_ = 5Å. This sequence has been repeated over a total of 20 h. This choice was motivated by the wish to get an accurate description of the time-evolution of the surface specific area. The sample holder was kept in a Peltier thermostat recirculating bath at the measurement temperature one hour before the insertion of the sample in the beamline. In the initial state, all the samples were kept at 25 °C (below cloud point) so that the sample temperature quickly “jumped” from 25 °C to the requested temperature, T_m_. 

All collected data have been treated by using Pasinet software for the isotropic gathering and to remove the noise and the intensity due to the empty cell. Treated data have been fitted by using the different equations presented in “Results and Discussion” by the minimization of χ^2^. 

## 3. Results and Discussion

### 3.1. Phase Diagram Construction

Solutions of polymer exhibiting a LCST are transparent below the cloud point and opaque above, due to the collapse and aggregation of polymer chains. Therefore, the measurements of the transmittance of PMAA solutions lead to a composition-temperature phase diagram of PMAA in D_2_O as a function of temperature. As shown in Figure 2a, the transmittance falls drastically above a critical temperature, the so-called cloud point temperature, T_cp._ This temperature decreases as the PMAA concentration is raised in the range of concentration studied (Φ_PMAA_ from 0.005 to 0.134). Higher concentrations were not considered for practical reasons related to gelation and issues in the temperature homogenization of the samples. Cloud points have been determined as the temperature corresponding to a transmittance of 0.5. Results are collected and depicted as blue squares in Figure 2b.

In order to complete this phase diagram and help the interpretation of SANS data, binodal points have been determined by studying the steady states reached under phase separation above the cloud point (see details in the Experimental Section). Solutions in the semi-dilute regime [47] (Φ_PMAA_ = 0.017, 0.065 and 0.104) were heated at fixed temperature above the cloud point. After 3 weeks, a concentrated phase at the bottom of the tube and a dilute phase at the top were observed. The PMAA volume fraction in each phase was characterized. The corresponding results are reported in Table 2 as well as on the phase diagram reported in Figure 2b; as explained in the experimental section, each phase separation experiment performed at a temperature T provides two points of volume fractions *Φ_PMAA,dil_*(T) and *Φ_PMAA,conc_*(T), displayed as green circles in Figure 2b. Indeed, these two steady-state phases correspond to the limits of stability at the temperature T and belong to the bimodal curve.

Cloud points determined by transmittance and binodal points obtained by the phase separation method display a very good agreement at low concentrations, where both experimental approaches could be reached. Such a good match between the cloud point curve and the binodal curve should be the sign of a quasi-binodal phase diagram [48]. Indeed, the PMAA homopolymer considered displays a rather low polydispersity index (1.25, see Table 1). Assuming a complete matching between these two curves allows us to obtain a binodal curve describing the phase behavior of PMAA for both low and high concentration regimes. 

This is the first time that such a complete phase diagram is obtained for PMAA/D_2_O. One should note that the fraction of the volume occupied by the polymer-rich phase, *x_conc_*, determined experimentally from the volumes of the two phases is in good agreement with the value extrapolated from the initial PMAA concentration (Φ_PMAA,0_) and the volume fractions measured for the two phases (*Φ_PMAA,dil_* and *Φ_PMAA,conc_*). Indeed, *x_conc_* should be equal to (Φ_PMAA,0_ − *Φ_PMAA,dil_*)/(*Φ_PMAA,conc_* + *Φ_PMAA,dil_*), as observed (see Table 2). This good agreement shows the quality of the different measurements. 

The LCST temperature can thus be estimated as the temperature corresponding to the minimum of the binodal curve (shown in Figure 2b) which corresponds to 48 °C in this case. This result is consistent with the value of the θ temperature in D_2_O determined by light scattering for dilute PMAA solutions, reported to be slightly below 50 °C [43].

Figure 2b also shows that the critical volume fraction ϕ_c_ associated with the LCST temperature is around 0.16. This volume fraction can be compared to that estimated from the Flory–Huggins model: (ϕ_c_)_FH_ = 1/(1 + λ^−1/2^ N_w_^1/2^), where N_w_ is the weight-average degree of polymerization and λ is a parameter that accounts for the polydispersity of the polymer sample (λ= N_z_/N_w_, N_z_ being the z-average degree of polymerization)_._ Assuming λ ≈ PDI (polydispersity index, see Table 1), (ϕ_c_)_FH_ may be estimated to be 0.016, which is much lower than the critical volume fraction determined in this work. Variations of the interaction parameter χ not only with temperature but also with the volume fraction are usually proposed to account for these deviations from the Flory–Huggins model [49,50]. 

At this stage, it is interesting to note the differences between the PMAA and PNIPAM phase diagrams: the polymer concentration in the concentrated phase is indeed much lower for PMAA than for PNIPAM. Such a change may result from distinct phase separation processes and lead to different structures for the resulting phases.

For this reason, the structures formed by PMAA solutions below and above the cloud point are investigated below.

### 3.2. Structure of the PMAA Solutions below the LCST

SANS measurements have been performed for PMAA solutions in D_2_O at 25 °C, below the cloud point temperature, for different polymer volume fractions in the semi-dilute regime, from Φ_PMAA_ = 0.0172 to 0.104. As previously explained, Φ_PMAA_*, the overlap volume fraction, has been determined by SAXS measurements carried out in the dilute regime and found to be equal to 0.0178.

The evolution of the scattered intensity, I, normalized by PMAA volume fraction, as a function of q is reported in Figure 3. It is clearly observed that for each concentration, the scattered intensity at 25 °C varies as *q*^−2^ at high *q* (above 4 × 10^−2^ Å^−1^, slope −2 in log-log scale). The curves *I*(*q*) are well-fitted to the Ornstein–Zernike relationship, usually used to describe semi-dilute polymer solutions: (2)$$I\left(q\right)=\frac{1}{1+q^{2}\xi^{2}}{\left(\rho_{PMAA}-\rho_{D_{2}O}\right)}^{2}Ioz\left(q=0\right)$$
in which $\rho_{PMAA}=2.33\times 10^{-6}\;{Å}^{2}$ and $\rho_{D_{2}O}=6.33\times 10^{-6}\;{Å}^{2}$ are the scattering length densities of PMAA and deuterated water. *Ioz*(*q* = *0*) stands for the scattered intensity extrapolated to *q* = 0 and normalized by the contrast. 

These fits lead to the determination of ξ, the correlation length of homogeneous polymer solutions in the semi-dilute regime. The evolution of ξ with the PMAA concentration is reported in Figure 3b. The dependence of ξ with Φ_PMAA_ is well-described by a power law: ${\xi \;\alpha \;\Phi }^{-a}$ with a = 0.53 (red line in Figure 3b). The exponent of the power law is related to the Flory exponent ν related to the solvent quality according to the relationship a = ν/(3ν − 1). The exponent, a, should be equal to 0.76 for a polymer in good solvent and to 1 in a θ solvent [51]. Here, the exponent (0.53) is closer to that predicted for polymers in good solvents than for θ solvents. This result suggests that PMAA in semi-dilute and concentrated solutions tends to behave as a polymer in good solvent at 25 °C.

This last result seems in apparent contradiction with previous results reported in the literature for dilute solutions of PMAA in water at a low degree of ionization. Indeed, some studies report that PMAA in water behaves like a polymer in poor solvent [32,52,53], due to the presence of the methyl group on the main chain. Besides, measurements by time-resolved fluorescence [33] indicate that PMAA chains display a hypercoiled structure at low ionization rate. However, PMAA displays a quite complex behavior as this hypercoiled conformation is characterized by the second virial coefficient, A_2_, reported to be equal to 0 at 30 °C [54] and follows a Mark–Houwink law with an exponent of 0.5, which is usually typical of polymers under the θ conditions. In the present case, the SANS experiments have been performed in the semi-dilute regime which has not yet been extensively explored in the literature. It is currently admitted that in this regime, the interaction parameter χ depends on the polymer volume fraction, a feature which is used to rationalize the deviation of the binary phase diagram from the predictions of the Flory–Huggins model (see above). In summary, it is shown in the present work that the inner structure of semi-dilute PMAA solutions at 25 °C reflects an interaction parameter close to the ones of polymers in a good solvent. 

### 3.3. Kinetics of the Phase Separation

As seen in the previous section, for a given polymer volume fraction, the PMAA solution in water phase separates as the temperature is increased above its cloud point. It is well known that the exact nature of such a transition is strongly related to the temperature quench depth. For instance, if the polymer solution undergoes a large temperature jump, a mechanism of spinodal decomposition mechanism is expected whereas a mechanism of nucleation and growth should preferentially occur following a low temperature jump. In order to get a deeper insight into the phase separation mechanism of PMAA in water and the structures formed above the LCST, time-dependent SANS measurements have been performed by using a small temperature quench (1 °C above the cloud point) and a larger one (10 °C above the cloud point).

#### 3.3.1. Structures of a Semi-Dilute Solution of PMAA in D_2_O +1 °C above Cloud Point (52 °C)

Time-dependent SANS experiments have been performed for a PMAA solution at Φ_PMAA_ = 0.065 at 52 °C, that is, 1 °C above the cloud point determined by transmittance measurements. As previously explained in the experimental section, the scattered intensity *I*(*q*) has been recorded every ten minutes for 12 h. For the sake of clarity only the curves obtained every 120 min have been included in Figure 4. Incoherent scattered intensity and the background have been subtracted from the raw data. The *I*(*q*) curve determined with the same sample at 25 °C has been included in the plot in order to highlight the consequences of the LCST transition. First, it is worth noting that at the length scale probed by these experiments, the phase separation is remarkably slow: the sample nanostructure still evolves after 12 h. In the case of PNIPAM, for instance, this transition takes less than one hour to reach a morphology with a similar surface-to-volume ratio [55]. However, after this first step, the formation of PNIPAM aggregates goes on. It should be remarked that the phase separation leads to objects in the colloidal domain for PNIPAM whereas an evolution of the phases is observed at the macroscopic scale in the case of PMAA.

As explained, at 25 °C, the scattered intensity *I*(*q*) displays a plateau at low *q* and a *q*^−2^ dependence at high *q* values (above 4 × 10^−3^ Å^−1^). The data have been fitted to the Ornstein–Zernike expression (Equation (2)). Above the cloud point temperature, the scattered intensity exhibits an increase at low *q* (below 0.015 Å^−1^) whereas at higher *q*, the *I*(*q*) curves follow the same power law as at room temperature. At low *q*, *I*(*q*) is now proportional to *q*^−4^. This power law indicates the formation of polymer-rich and polymer-poor domains, the interfaces of which lead to strong forward scattering described by a Porod law [56]. Each domain is so large that their characteristic size is not accessible by SANS. Based on these features, the SANS intensity curves *I*(*q*) have been analyzed using Equation (3), which corresponds to the sum of the scattering function for dense polymer globules and of the Orstein–Zernike function:(3)$$I\left(q\right)=\frac{{\left(\rho_{PMAA}-\rho_{D_{2}O}\right)}^{2}Ioz\left(q=0\right)}{1+q^{2}\xi^{2}}+x_{conc}\frac{2\;\pi {\left(\rho_{conc}-\rho_{dil}\right)}^{2}}{q^{4}}.\;\frac{S}{V}$$
where *S*/*V* stands for the interface area per unit volume of the PMAA-rich phase, *x_conc_* corresponds to the sample volume fraction it occupies, *ρ_conc_* and *ρ_dil_* are the neutron scattering length density of the concentrated and dilute phases, respectively.

In order to analyze the SANS data using Equation (3), the polymer volume fraction for each phase (*Φ_PMAA,conc_* and *Φ_PMAA,dil_*) and the sample volume fraction occupied by the polymer-rich phase ($x_{conc}$) have been determined as detailed in the experimental section. The results are reported in Table 2. The scattering length density of both concentrated and dilute phases has been calculated using Equations (4a) and (4b).
(4a)$$\rho_{conc}=\Phi_{PMAA,conc}\rho_{PMAA}+\left(1-\Phi_{PMAA,conc}\right)\rho_{D2O}$$
(4b)$$\rho_{dil}=\Phi_{PMAA,dil}\rho_{PMAA}+\left(1-\Phi_{PMAA,cdil}\right)\rho_{D2O}$$

The values obtained are also included in Table 2. In most of the contributions investigating the phase transition of PNIPAM by SANS, the scattering length densities have been approximated to that of the pure polymer for the polymer-rich phase and to that of the solvent for the dilute phase [45,55]. Such an approximation is valid for PNIPAM since the volume fraction of polymer in the dense globules is very high [21,57,58,59,60]: the correlation of SANS measurements and Diffusive Wave Spectroscopy (DWS) [45] leads to a volume fraction of PNIPAM ranging between 90 and 100% in the concentrated phase. This situation strongly differs to the case of PMAA: the polymer fraction in PMAA-rich domains is comparatively much lower, which implies that the approximation concerning the scattering length density would not allow a robust analysis of the SANS data. At Φ_PMAA_ = 0.065 and +1 °C above the cloud point, this volume faction amounts to *Φ_PMAA,conc_* = 0.16. This means that the scattering length density for this phase is closer to that of deuterated water than to that of pure PMAA. 

The fit of all the experimental *I*(*q*) curves measured over time to Equation (3) allows us to reconstruct the time evolution of ξ, the apparent correlation length during the phase separation (Figure 5). Due to inhomogeneities within the samples between the polymer-rich and the polymer-poor phases, this correlation length is less easy to interpret than in a homogeneous polymer solution. In a first approach, it can nevertheless be qualitatively interpreted as the correlation length of the polymer rich phase even though this value is probably overestimated. In Figure 5, it is observed that ξ decreases with time. After 12 × 10^3^ s (more than 3 h), ξ reaches a plateau value around 26 Å which is assigned to the collapse of the PMAA chains and to their concentration in the polymer-rich phase. In comparison to the results obtained at 25 °C, such correlation length corresponds to a highly concentrated solution. Based on the PMAA volume fraction measured at this temperature in the concentrated phase, *Φ_PMAA,conc_* = 0.20, the ξ value which would be obtained in a single-phase state was estimated using the power law fit at 25 °C (red line on Figure 3b) would be equal to 19.5 Å. The difference between this value and that determined by SANS measurement at 52 °C may originate from two different features: first, the change of the solvent quality of PMAA between 25 °C et 52 °C (as the temperature increases the solvent quality may be decreased and the power law at 25 °C is probably not valid anymore); second as mentioned earlier, the value of ξ measured at 52 °C is overestimated because there is a contribution from both dense and dilute phases to *I*(*q*). In the literature, an estimation of ξ above the cloud point for a semi-dilute solution of PNIPAM was reported by Meier–Koll et al. [55]. A decrease of the correlation length above the cloud point has also been observed. However, in their contribution, this value has been obtained at a fixed time (20 min) after a temperature jump where they expect to have reached the plateau value. 

Another interesting parameter deduced from the fit of the SANS data to Equation (3) is the specific surface between the two phases formed during the phase separation. The time evolution of (*S*/*V*), shown in Figure 6, provides an insight on both the structures formed and the processes occurring during the LCST phase separation. Figure 6 clearly indicates that two different behaviors occur above the cloud point (T_CP_ + 1 °C). During the first 8 × 10^3^ s (more than 2 h), the time evolution of S/V may be satisfactorily described using a power law: *S*/*V*α t^−0.3^. The exponent −0.3 is close to that obtained for an Ostwald ripening mechanism [61,62] of the growth of droplets. Such a mechanism is generally expected in the first stage of a nucleation-growth (NG) process [62,63]. After this first regime, the data of Figure 6 may by fitted to a different power law with a higher exponent, *S*/*V* α t^−1.6^. Several mechanisms of droplet growth have been reported in the literature. An Ostwald ripening mechanism with an exponent 1/3 for the power law describing the time evolution of the radius of the remaining droplets is reported for several polymer/solvent binary mixtures (hydroxypropyl cellulose/water [64], polystyrene/cyclohexanol [65], methylcellulose/NaCl/water [66]). A viscoelastic phase separation process (VPS) has also been described by Tanaka for asymmetric mixtures [67]. This dynamic asymmetry can be induced by different viscoelasticity for each phase: this may be the case for PMAA-rich and PMAA-poor phases, which should exhibit a different viscosity. Lastly, a growth of the solvent-poor domains controlled by hydrodynamic process with a domain size following a power law characterized by an exponent 1 is reported in several cases: polystyrene/cyclohexanol after the stage of nucleation and growth [65], PNIPAM/water [55].

#### 3.3.2. Structures of a Semi-Dilute Solution of PMAA in D_2_O + 10 °C above Cloud Point (61 °C)

The same kind of measurements have been performed 10 °C above the cloud point temperature (61 °C) (Appendix A). The data have been analyzed as previously explained which allows us to monitor both the time evolutions of ξ and of S/V while the phase transition occurs. As for the LCST phase separation at T_CP_ +1 °C, a strong decrease of both quantities with time is observed (Figure 5 and Figure 6). At 10 °C above the cloud point, the transition is faster than at T_CP_ + 1 °C. Indeed, the plateau value of ξ vs. time is already reached between the first and the second measurement. The plateau value of ξ now reaches 20 Å whereas the value of this plateau was around 26 Å, 1 °C above cloud point (CP). Moreover, it is again higher than the value (16 Å) obtained using the power law ξ α Φ^−0.53^ derived below the cloud point (at 25 °C). The volume fraction of PMAA in the concentrated phase when the phase separation occurs at T_CP_ + 10 °C is *Φ_PMAA,conc_* = 0.26. It should be noted that the evolution of ξ is thus ξ consistent: ξ is indeed expected to decrease as the PMAA volume fraction in the concentrated phase increases, while *Φ_PMAA,con_*_c_ at T_CP_ + 10 °C was found to be higher than at T_CP_ + 1 °C (0,26 versus 0.20, see Table 2). The overall change of the time evolution of ξ over the LCST phase separation upon raising the temperature at which this latter occurs is furthermore consistent with that reported for PNIPAM [55,68].

At T_CP_ + 10 °C, the S/V ratio also decreases with time and as for T_CP_ + 1 °C, two different regimes can be detected over the probed time scale. Over a long time (beyond 8.10^3^ s) S/V exhibits a time evolution that can be described using a power law, the exponent of which is very similar to that determined at T_CP_ + 1 °C (−1.44 versus −1.6). By contrast, for short times, the exponent of the power law that describes the time variation of S/V significantly differs from that observed at T_CP_ + 1°: −0.58 here against −0.3 (T_CP_ + 1 °C). This evolution could be interpreted as the growing of a spinodal phase-separation. 

#### 3.3.3. Effect of the PMAA Concentration

Similar SANS measurements have been performed at two additional PMAA volume fractions: the one at the lower limit of the semi-dilute regime (Φ_PMAA_ = 0.017) and the other at the upper limit below which a homogeneous sample can be prepared (Φ_PMAA_ = 0.104) (Appendix A). A temperature jump from 25 °C up to 10 °C above the cloud point temperature measured at 50% transmittance (69 °C for Φ_PMAA,0_ = 0.017 and 59 °C for Φ_PMAA,0_ = 0.104) has been applied and scattered intensities *I*(*q*) were collected as a function of time every 10 min. As before, the intensity at high q is proportional to *q*^−2^ whereas a Porod law (*I*(*q*) α *q*^−4^) is observed at small q. In time, the portion of the *I*(*q*) curve displaying a q^−4^ behavior is shifted towards low q as the time increases. This trend indicates that the size of PMAA-rich domains increases. SANS results have been analyzed as previously explained and the time-evolution of the apparent correlation length ξ and of the specific area (which correspond to S/V divided by the PMAA concentration) is shown in Figure 7a,b, respectively.

For all the concentrations studied, the plateau value of ξ vs. time is reached quickly after the temperature jump. As expected from our previous observations, this plateau value decreases as the initial PMAA volume fraction is raised from 0.065 to 0.104. It may at first seem surprising that the plateau value of ξ for the lowest concentration (Φ_PMAA,0_ = 0.017) is lower than that measured for Φ_PMAA,0_ = 0.104. This result is nevertheless consistent with the values of the concentration of the PMAA-rich phase obtained following the phase separation at 69 °C. Indeed, under these conditions, *Φ_PMAA,conc_* = 0.265 at 69 °C for Φ_PMAA,0_ = 0.017 (see Table 2), whereas *Φ_PMAA,conc_* = 0.23 when Φ_PMAA,0_ = 0.104. The measured correlation length ξ roughly corresponds to the mesh size of the polymer-rich domains and as a result, decreases as the concentration in this phase increases.

Finally, for the two additional Φ_PMAA,0_ values, the specific area decay during the phase separation exhibits regimes that can be described again by two distinct power laws. The exponents of these power laws are reported in Table 3. This is discussed further below.

### 3.4. Discussion of the Time Decays of S/V

The specific surface S/V provides an estimate of the size of the phase-separated domains. If one assumes that concentrated domains are spherical droplets of radius R, then *S*/*V* = 3 *x_conc_*/R [45]. The time evolution of S/V is thus a direct probe of the kinetics of the PMAA-rich domain growth. Considering that large-size domains are already formed right at the beginning of the measurements (S/V is around 100 mm^−1^ after 1000 s), one may question the role of gravity in the phase-separation mechanism. For this purpose, it is interesting to compare the time t_g_ required for droplets of a radius R to fall down through a distance equal to one diameter (2R) to the time t_D_ needed for the droplets to diffuse over the same distance by experiencing Brownian motion.
(5a)$$t_{g}=2R/[2R^{2}\;(\rho_{conc}-\rho_{dil})\;g/(9\eta )]$$
t_d_ = (2R)^2^ 6π η R/(k_b_ T)(5b)
where ρ*_conc_* and ρ*_dil_* are the densities of the concentrated and dilute phases respectively. η is the viscosity of the dilute phase at the considered temperature T. Calculations were performed for the PMAA solution in water with an initial polymer volume fraction Φ_PMAA,0_ of 0.065. When this latter is heated from 25 °C to 1 °C above the cloud point temperature (52 °C), phase-separation occurs and lead to a PMAA volume fraction in the polymer-rich domains *Φ_PMAA,conc_* of 0.20. At 1000 s, R = 8 μm. It has been assumed that at this temperature T_CP_ + 1 °C = 52 °C, ρ*_conc_* − ρ*_dil_* is the same as that measured at 25 °C and that the viscosity of the dilute phase is close to that of the pure solvent. Under these conditions time t_g_ (2.5 s) is found to be much lower than t_d_ (5600 s). The same order of magnitude was found for t_g_ and t_d_ for all the other samples studied in the semi-dilute range. The fact that t_g_ << t_d_ implies that sedimentation should be much faster that Brownian dynamics. However, if gravity was fully governing the phase separation process, this latter should be complete within a few hours whereas in the present case, it has been observed that demixing occurs after several weeks, for all the semi-dilute PMAA samples. In fact, the sedimentation of the droplets should be arrested in these systems due to the formation of a percolating network of droplets. The PMAA concentration within the droplets being quite high (PMAA volume fractions from 0.20 to 0.27, see Table 2), the concentrated domains should behave as microgels. It has indeed been shown that PMAA solutions with volume fractions higher than 0.077 exhibit a rheological gel-like behavior [27,47]. As soon as the microgel-like domains form a connected network, the whole sample gets solid-like at a macroscopic length scale and the sedimentation is then frozen. Indeed, the PMAA concentration before the LCST phase separation (Φ_PMAA,0_) should be higher than a critical value to allow the formation of a percolating network. For instance, it has been reported that the critical volume fraction for the percolation of randomly placed spheres is about 0.18 [69]. This value lies in the same range as the *x_conc_* values reported in Table 2 for the PMAA solutions in the semi-dilute range. During this time, for dilute PMAA solutions, phase separation should occur above the cloud point temperature and the resulting *x_conc_* value should be comparatively lower than that resulting from phase separation of semi-dilute PMAA samples. For a solution with Φ_PMAA,0_ = 0.017, *x_conc_* was found to be equal to 0.05, to be compared to 0.21 and 0.43 obtained for Φ_PMAA,0_ = 0.065 and Φ_PMAA,0_ = 0.104, respectively. Such a low *x_conc_* should not allow the formation of a percolating network of droplets. Therefore, in that case, their phase separation should be controlled by gravity. In good agreement with this expectation, it has been checked that PMAA solutions prepared in the dilute regime exhibit macroscopic phase separation in a few hours while semi-dilute solutions remain homogeneously turbid during a period of the same duration (see photographs in Appendix A). All these features provide corroborative evidence that sedimentation is arrested in the case of PMAA solutions in the semi-dilute regime due to the formation of a percolating network of microgel-like droplets.

Figure 6 and Table 3 show that S/V decreases as a function of time during the LCST phase separation. This trend results from a growth of the droplet size even though the sedimentation process is arrested. The phase-separating PMAA solutions display two distinct regimes of droplet growth: a first regime occurring over a relatively short time scale, with power law exponents between 0.30 and 0.65 for the evolution of *S*/*V*, and a second regime with exponents in the range 1.40–1.70. This second regime is characterized by higher power law exponents than those usually reported. Such a feature could result from a non-isotropic growth of the droplets. Indeed, when the droplets grow, their weight is increasing, making the network more and more prone to deformation or collapse under gravity. Indeed, the maximum gravity stress σ_g_ exerted on a single droplet of radius R is of order:(6)$$\sigma_{g}=(R/3)\;\times \;(\rho_{conc}-\rho_{dil})\;\times \;g$$

This stress increases with the radius R of the droplet. When it reaches the critical stress of the macroscopic network, the droplets growth should be non-isotropic due to the influence of gravity. Fracture or elastic deformation could occur. It should be noted that the crossover between the two regimes of growth occurs at a radius R ~ 25 μm and is similar for the different experiments (see Figure 6 and Figure 7); this supports the occurrence of a critical stress above which the droplet growth becomes anisotropic. A similar crossover between two regimes has been reported for PNIPAM/water by Meier Knoll [58], with a power law exponent in the second regime (1.89) quite close to those obtained in the present work. In contrast, the droplet growth is expected to be isotropic in the first regime corresponding to the short time scale. This process has been schematized in Figure 8. The exponent 0.30 determined in the case of the experiment carried out with a low temperature jump (1 °C above the cloud point) is clearly in agreement with an Ostwald ripening mechanism. The experiments performed with a larger temperature quench (+10 °C above the cloud point) evidence faster growth mechanisms of the PMAA-rich domains. The exponent of the power laws that satisfactorily describe the time-evolution of S/V neither corresponds to an Ostwald ripening mechanism nor to a hydrodynamic process. Therefore, one may consider that under these conditions, the LCST transition involves a viscoelastic phase separation process. Indeed, the asymmetry (and in particular, the viscosity difference) between both concentrated and dilute phases might be enhanced in the case of a large temperature jump as suggested by the more pronounced difference between *Φ_PMAA,conc_* and *Φ_PMAA,dil_* (see Table 2).

## 4. Conclusions

The present work investigates the phase transition of a LCST polymer, PMAA, in the semi-dilute regime. A temperature-composition phase diagram has been constructed for the first time, using transmittance measurements but also the analysis of phase-separated samples, allowing an accurate determination of the LCST temperature of PMAA in D_2_O (48 °C). In a second step, the phase microstructures formed during the phase transition of PMAA have been studied by SANS. While transmittance measurements show instantaneous changes at the cloud point temperature, SANS shows that the structures evolve during more than 20 h after the temperature jump above the cloud point. An original mechanism of phase separation is evidenced in the semi-dilute regime. The phase separation occurs through the formation of microgel-like droplets. The formation of a percolating network of droplets arrests the influence of gravity, thus accounting for the slow kinetics of PMAA phase separation in the semi-dilute regime. 

## Figures and Tables

**Figure 1 polymers-14-04708-f001:**
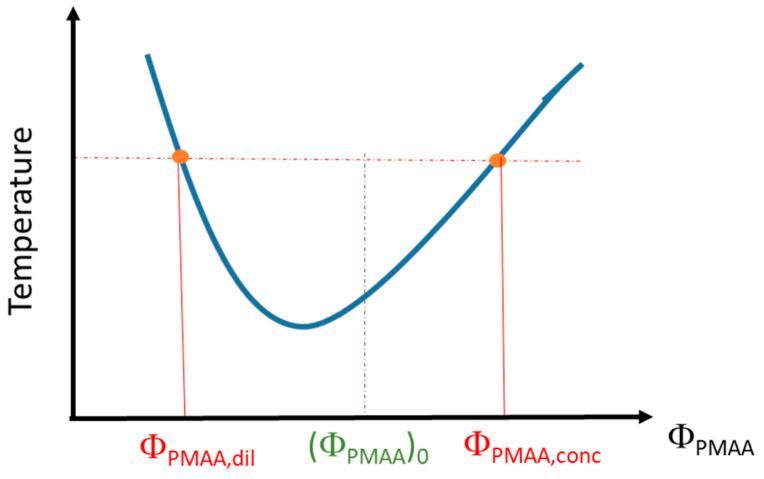
Scheme of the phase diagram determination from the binodal curve.

**Figure 2 polymers-14-04708-f002:**
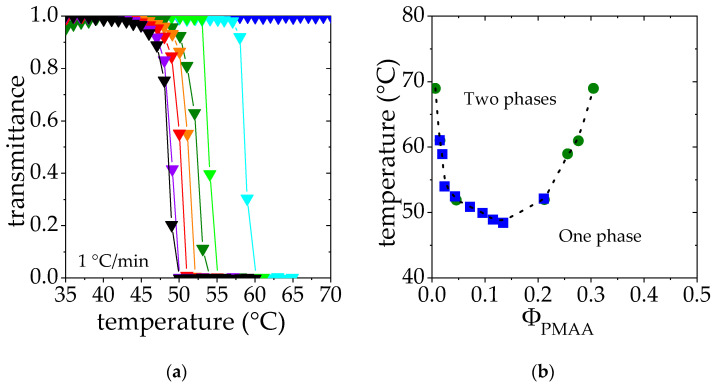
(**a**). Transmittance measurements of PMAA solution at different PMAA volume fraction Φ_PMAA_: ▼ 0.005, ▼0.019, ▼0.028, ▼0.047, ▼0.075, ▼0.0950, ▼0.115, ▼0.134. (**b**). Cloud point temperature as a function of the PMAA volume fraction determined through transmittance measurements ■. These data are superimposed on the phase diagram derived from the phase concentration measurements ●. Dashed line is a “guide for the eyes”.

**Figure 3 polymers-14-04708-f003:**
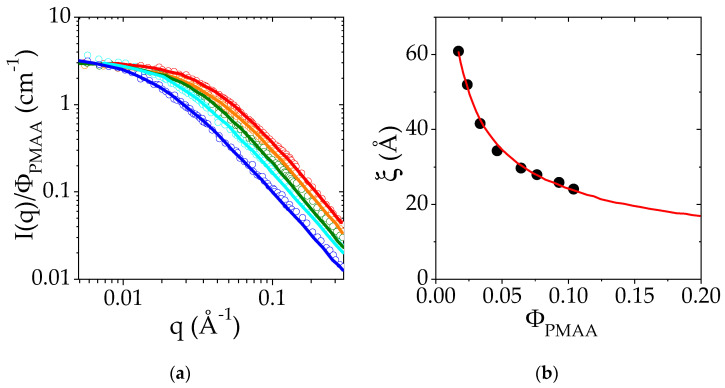
(**a**) Small-Angle Neutron Scattering intensity I as a function of the scattering wavevector q measured at 25 °C for different PMAA volume fractions in the semi-dilute regime (from 0.023 to 0.104): Φ_PMAA_ = 0.023 in dark blue, Φ_PMAA_ = 0.038 in light blue, Φ_PMAA_ = 0.058 in green, Φ_PMAA_ = 0.0775 in yellow, Φ_PMAA_ = 0.104 in red. The *I*(*q*) curves were divided for the sake of comparison by Φ_PMAA_. Solid lines are the fits of the experimental data to an Orstein–Zernike function. (**b**) Variation of the correlation length ξ with the polymer volume fraction, Φ_PMAA_. The red line corresponds to the fit of ξ (Φ_PMAA_) to a power law.

**Figure 4 polymers-14-04708-f004:**
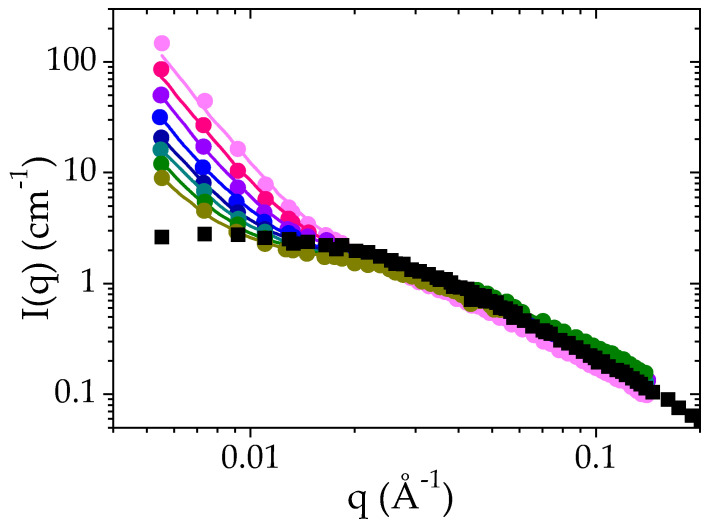
Time evolution of the SANS intensity as a function of q for a solution of PMAA at a volume fraction of 0.065 after a temperature jump from 25 °C to T_cp_ + 1 °C: 60 min ●, 180 min ●, 300 min ●, 420 min ●, 520 min ●, 660 min ●, 780 min ● and 900 min ●. solid lines correspond to the fit of the experimental data to Equation (3). The *I*(*q*) curve recorded for the same solution at 25 °C is also plotted for the sake of comparison (■).

**Figure 5 polymers-14-04708-f005:**
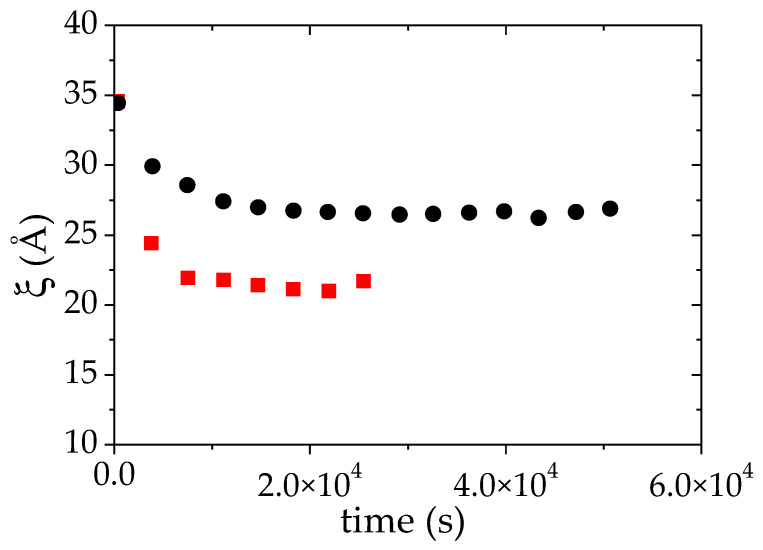
Evolution of the apparent correlation length ξ during the LCST phase separation of PMAA solution in D_2_O (Φ_PMAA,0_ = 0.065), induced by a temperature jump from 25 °C up to: +1 °C (●) and +10 °C (■) above the cloud point. ξ was deduced from the fit of the SANS data to Equation (3).

**Figure 6 polymers-14-04708-f006:**
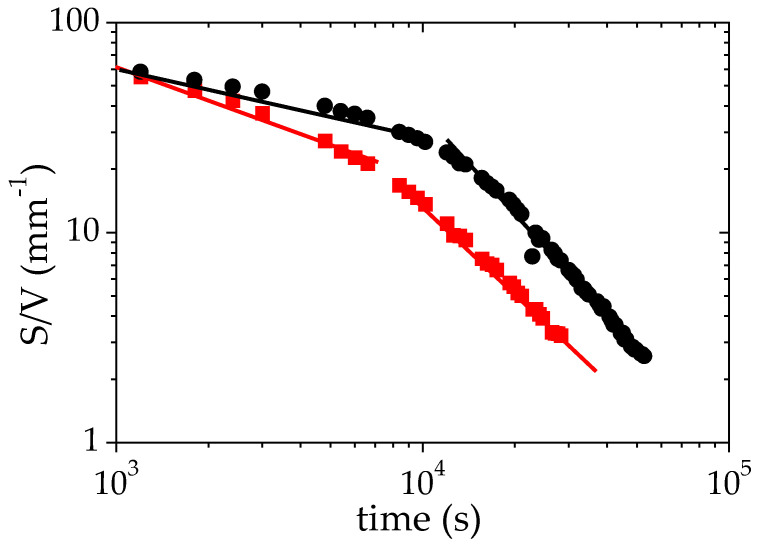
Variation of the specific surface S/V as a function of time during the phase separation of a solution of PMAA in D_2_O (Φ_PMAA,0_ = 0.065) under isothermal conditions +1 °C (●) and +10 °C (■) above the cloud point. The solid lines correspond to the fits to power laws for the two regimes identified on this plot.

**Figure 7 polymers-14-04708-f007:**
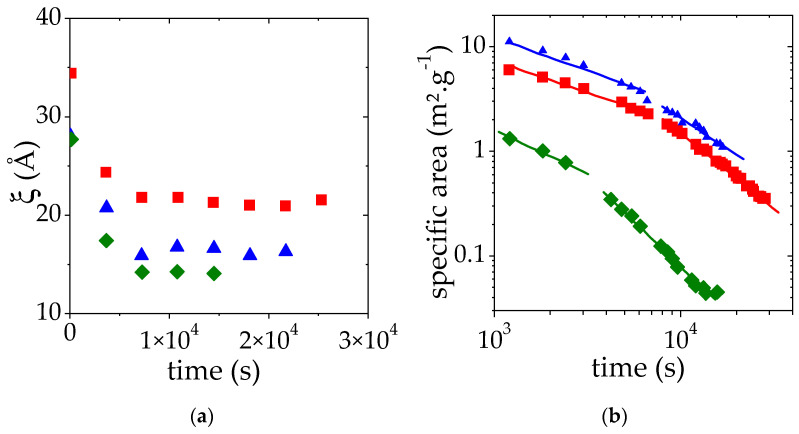
Apparent correlation length obtained ξ (**a**) and specific area (**b**) as a function of time for a solution at a volume fraction of 0.017 (♦), 0.065 (■), 0.104 (▲). This variation was monitored during the LCST phase separation induced by a temperature jump from 25 °C to T_CP_ + 10 °C. ξ and specific area were deduced from the fit of SANS data by Equation (3). The solid lines correspond to the fit of the time evolution of the specific area to power laws.

**Figure 8 polymers-14-04708-f008:**
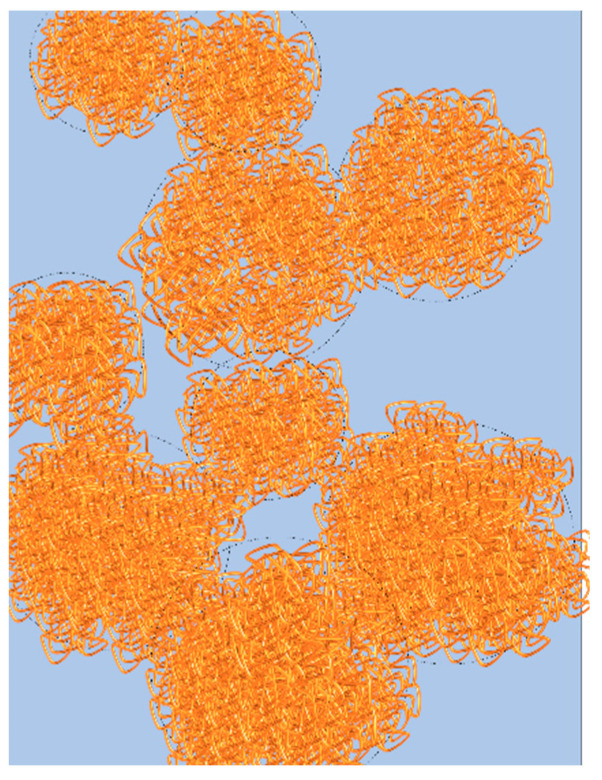
Graphical representation of the percolated network of droplets. Polymer rich phase in represented in orange and polymer poor phase in blue.

**Table 1 polymers-14-04708-t001:** PMAA characteristics. Molar masses and PDI are determined by SEC, *R_g_* by SAXS and *c** by using Equation (1).

*M_w_* (g.mol^−1^)	M_n_ (g.mol^−1^)	PDI	*R_g_* (Å)	*c** (g/L)
391,000	313,000	1.25	190	23

**Table 2 polymers-14-04708-t002:** Volume fraction of PMAA in each phase above the cloud point for different PMAA concentrations and different temperatures. (Φ_PMAA_)_0_ stands for the initial PMAA volume fraction while *Φ_PMAA_,_dil_* and *Φ_PMAA_,_conc_* are the volume fraction in the dilute and concentrated phase. (x_conc_)_measured_ is equal to the ratio between the volume of the concentrated phase and the total volume of the sample. The scattering length densities ($\rho_{dil}$ and $\rho_{conc}$) of both dilute and concentrated phases were calculated on the basis of *Φ_PMAA_,_dil_* and *Φ_PMAA_,_conc_*.

(Φ_PMAA_)_0_	0.017	0.065	0.065	0.104
**Temperature T**	69 °C	52 °C	61 °C	59 °C
**Temperature above cloud point**	+10 °C	+1 °C	+10 °C	+10 °C
** *Φ_PMAA,dil_* **	0.0038	0.040	0.013	0.015
** *Φ_PMAA,conc_* **	0.265	0.200	0.260	0.230
**(x_conc_)_extrapolated_**	0.051	0.156	0.21	0.42
**(x_conc_)_measured_**	0.05	0.16	0.21	0.43
$\rho_{conc}\;$ **× 10^−6^ Å^−2^**	5.26	5.53	5.30	5.42
$\rho_{dil\;}$ **× 10^−6^ Å^−2^**	6.31	6.17	6.28	6.27

**Table 3 polymers-14-04708-t003:** Exponents β of the power laws describing the time evolution of *S*/*V* (*S*/*V* α t ^β^) occurring during the LCST phase separation of PMAA solutions at different concentrations. The transition is monitored following a temperature jump from 25 °C up to T_CP_ + 1 °C and T_CP_ + 10 °C).

(Φ_PMAA_)_0_		0.017	0.065	0.104
	**T-T_CP_**			
**β (short-time regime)**	1 10	- 0.65	0.30 0.58	- 0.63
**β (long-time regime)**	1 10	- 1.40	1.60 1.44	- 1.69

## Data Availability

The data presented in this study are available on request from the corresponding author.

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
