# Peer review of "Unexpected Slow Kinetics of Poly(Methacrylic Acid) Phase Separation in the Semi-Dilute Regime"

_polymers, 2022, doi:10.3390/polym14214708_

Round 1
Reviewer 1 Report
The manuscript by Robin et al. reports their investigation regarding the unexpected slow kinetics of PMAA phase separation in the semi-dilute regime. The author investigated the phase transition of an LCST polymer (PMAA) in the semi-dilute solution, and the authors mentioned that the nature of the aggregates formed in the semi-dilute regime was not previously elucidated. This finding is new and the presentation is concise and clear, and the conclusions have largely supported the calculations. Despite the large supporting data towards these findings, I feel the manuscript requires some broader outlook and immediate significance to express the value of this work to a broader audience. Thus I recommend this paper for publication after addressing the following two questions.
1. What is the major advance reported in this manuscript?
2. What will be the immediate significance of this advance?
Author Response
We thank referees for their comments and suggestions.
1.There are two main advances in this paper:
_ The first complete phase diagram of a PMAA which is an often used polymer but quiet unknown at high temperature.
_A complete study of the structure of PMAA semi diluted solutions above cloud point.
2.
The immediate significant advance of this study is that PMAA microstructuring can be controlled through the process of phase separation and could be used to prepare well defined porous PMAA networks.
Reviewer 2 Report
This manuscript reports on the impact of slow kinetics of poly(methacrylic acid) phase separation in the semi-dilute regime. The topic is interesting and the conclusions seem sound.
However, this work is a continuation of the previous study: Robin, C., Lorthioir, C., Amiel, C., Fall, A., Ovarlez, G., & Le Cœur, C. (2017). Unexpected Rheological Behavior of Concentrated Poly(methacrylic acid) Aqueous Solutions. Macromolecules, 50(2), 700–710. doi:10.1021/acs.macromol.6b01552.
The article is well structured and organized, and easy to read. The main ideas and concepts are undoubtedly logically structured and focused on the topic. The figures are clear, well presented, and help the reader to follow the point.
I would like to especially note a detail description of the experimental procedures and mathematical methods for the processing of the obtained data.
I would suggest a scheme of the described transformations of PMAA solutions since it will enrich the presented paper.
In addition, to obtain experimental data, the authors used the Small-Angle Neutron Scattering (SANS) method. Are there examples of works where other methods were employed to conduct similar studies? If yes, the authors should cite such papers.
Author Response
I would like to especially note a detail description of the experimental procedures and mathematical methods for the processing of the obtained data.
Data treatment and fitting procedure have been added in the article (line 171-173)
I would suggest a scheme of the described transformations of PMAA solutions since it will enrich the presented paper.
A figure has been added.
In addition, to obtain experimental data, the authors used the Small-Angle Neutron Scattering (SANS) method. Are there examples of works where other methods were employed to conduct similar studies? If yes, the authors should cite such papers.
A paper of a kinetical study on PNIPAM by SANS has been cited (ref 58) . However, such kind of kinetical studies of LCST behavior is poorly addressed id literature. This is probably due to the fact that, for “classical” LCST-behavior polymer such as PNIPAM, the equilibrium is reached in less than 20 minutes.
